# Cardioprotective Role of BGP-15 in Ageing Zucker Diabetic Fatty Rat (ZDF) Model: Extended Mitochondrial Longevity

**DOI:** 10.3390/pharmaceutics14020226

**Published:** 2022-01-19

**Authors:** Mate Kozma, Mariann Bombicz, Balazs Varga, Daniel Priksz, Rudolf Gesztelyi, Vera Tarjanyi, Rita Kiss, Reka Szekeres, Barbara Takacs, Akos Menes, Jozsef Balla, Gyorgy Balla, Judit Szilvassy, Zoltan Szilvassy, Bela Juhasz

**Affiliations:** 1Department of Pharmacology and Pharmacotherapy, Faculty of Medicine, University of Debrecen, H-4032 Debrecen, Hungary; matekozma@yahoo.com (M.K.); bombicz.mariann@pharm.unideb.hu (M.B.); varga.balazs@pharm.unideb.hu (B.V.); priksz.daniel@pharm.unideb.hu (D.P.); gesztelyi.rudolf@pharm.unideb.hu (R.G.); veratarjanyi@gmail.com (V.T.); kiss.rita@med.unideb.hu (R.K.); szekeres.reka@med.unideb.hu (R.S.); takacs.barbara@pharm.unideb.hu (B.T.); menesaki@hotmail.com (A.M.); szilvassy.zoltan@med.unideb.hu (Z.S.); 2Institute of Internal Medicine, Faculty of Medicine, University of Debrecen, H-4032 Debrecen, Hungary; balla.jozsef@med.unideb.hu; 3Department of Paediatrics, Clinical Centre, University of Debrecen, H-4032 Debrecen, Hungary; balla@med.unideb.hu; 4Department of Oto-Rhino-Laryngology and Head and Neck Surgery, Faculty of Medicine, University of Debrecen, H-4032 Debrecen, Hungary; szj@med.unideb.hu

**Keywords:** ageing, diabetic cardiomyopathy, mitochondrial longevity, electron transport chain, BGP-15, nicotinic acid, NAD precursor, bioactive molecule, antioxidants

## Abstract

Impaired mitochondrial function is associated with several metabolic diseases and health conditions, including insulin resistance and type 2 diabetes (T2DM), as well as ageing. The close relationship between the above-mentioned diseases and cardiovascular disease (CVD) (diabetic cardiomyopathy and age-related cardiovascular diseases) has long been known. Mitochondria have a crucial role: they are a primary source of energy produced in the form of ATP via fatty acid oxidation, tricarboxylic acid (TCA) cycle, and electron transport chain (ETC), and ATP synthase acts as a key regulator of cardiomyocyte survival. Mitochondrial medicine has been increasingly discussed as a promising therapeutic approach in the treatment of CVD. It is well known that vitamin B3 as an NAD^+^ precursor exists in several forms, e.g., nicotinic acid (niacin) and nicotinamide (NAM). These cofactors are central to cellular homeostasis, mitochondrial respiration, ATP production, and reactive oxygen species generation and inhibition. Increasing evidence suggests that the nicotinic acid derivative BGP-15 ((3-piperidine-2-hydroxy-1-propyl)-nicotinic amidoxime) improves cardiac function by reducing the incidence of arrhythmias and improves diastolic function in different animal models. Our team has valid reasons to assume that these cardioprotective effects of BGP-15 are based on its NAD^+^ precursor property. Our hypothesis was supported by an animal experiment where ageing ZDF rats were treated with BGP-15 for one year. Haemodynamic variables were measured with echocardiography to detect diabetic cardiomyopathy (DbCM) and age-related CVD as well. In the ZDF group, advanced HF was diagnosed, whereas the BGP-15-treated ZDF group showed diastolic dysfunction only. The significant difference between the two groups was supported by post-mortem Haematoxylin and eosin (HE) and Masson’s trichrome staining of cardiac tissues. Moreover, our hypothesis was further confirmed by the significantly elevated Cytochrome c oxidase (MTCO) and ATP synthase activity and expression detected with ELISA and Western blot analysis. To the best of our knowledge, this is the first study to demonstrate the protective effect of BGP-15 on cardiac mitochondrial respiration in an ageing ZDF model.

## 1. Introduction

The close association between diabetes and cardiovascular disease has long been known. In addition to being a risk factor for coronary atherosclerosis and ischemic heart disease, diabetes also leads directly to myocardial dysfunction (diabetic cardiomyopathy) [1]. Metabolic abnormalities and the production of reactive oxygen species (ROS) can cause pathological activation of several signalling pathways, thus modifying the myocardial expression of various genes [2]. Ageing myocardial oxidative stress, myocardial hypertrophy, and fibrotic remodelling, along with increased apoptosis, all have a critical role in the decreased myocardial ability to contract and relax. Age-related defects in mitochondrial function have been related to normal cardiac ageing. It has been reported that the mitochondrial respiratory chain complexes decline with age in cardiac muscle, particularly in complex I and IV, although complexes II, III, and V are less affected by age in cardiomyocytes. It is widely accepted that nicotinamide adenine dinucleotide (NAD^+^) and its reduced and phosphorylated forms such as NADH, NADPH, and NADP^+^ are crucial regarding mitochondrial. Furthermore, NAD^+^ harmonizes the function of several pathways such as glycolysis, TCA cycle, oxidative phosphorylation (OXPHOS), or fatty acid and amino acid metabolism as a cofactor for a wide range of oxidoreductase reactions. NAD^+^ is converted to NADH in the mitochondrial compartment, and later it is oxidized by complex I, which is the first part of the mitochondrial electron transport chain [3]. Finally, ATP will be generated from ADP through a series of redox reactions (OXPHOS) [4]. NADP and NADPH have a prominent role in the fight against oxidative damage and the prevention of oxidative stress-induced damage, also under physiological conditions [5]. Vitamin B3 and its derivatives (nicotinic acid, nicotinamide, and nicotinamide riboside) are precursors of nicotinamide adenine dinucleotide (NAD), thus playing a key role in the above-mentioned processes. Several studies confirmed that, in high doses, niacin can be useful in hypercholesterolemia, mitochondrial myopathy or other systemic NAD^+^ deficiency [6]. The drug candidate BGP-15, also named N’ -(2-hydroxy-3-(piperidin-1-yl)propoxy)-3-pyridine-carboximidamide, is a nicotinic acid amide derivative, which is a particularly important feature considering the fact that nicotinic acid amide, which can also be called niacin amide or nicotinamide, is the water-soluble active form of vitamin B3 [7,8]. Several studies confirm the beneficial effects of BGP-15, including its impact on mitochondrial and cardiac function especially in animal models [9,10]. There is increasing evidence that BGP-15 improves muscle strength and function and can restore cardiac function in animal models of muscular dystrophy, heart failure, and diabetic cardiomyopathy [11,12]. In a previous study, our research group established that the drug candidate improves diastolic function in Goto–Kakizaki rats suffering from diabetic cardiomyopathy [13]. Additionally, there is further evidence for the beneficial effect of BGP-15 on mitochondrial functions exerted via the reduction of reactive oxygen species (ROS), which indicates the key role of BGP-15 in the prevention of oxidative damage, although these studies are confined to cell cultures [14]. Although there are several animal models that provide a great opportunity for researchers to examine diabetes mellitus or cardiac diseases, we need to emphasize that the Zucker Diabetic Fatty rat is one of the most representative experimental models that can exhibit these two features. The aim of the present study was to confirm the outstanding importance of drug candidate BGP-15 in enhancing mitochondrial function in this ageing ZDF animal model. Last but not least, we also examined the subsequent effects on the heart function in the aforementioned ageing type 2 diabetes mellitus rat model.

## 2. Materials and Methods

### 2.1. Animal Model

In the present study, 30 ZDF male rats (8-week-old, 200 g), and 10 LEAN male rats (8-week-old, 200 g) as control group were used, which were purchased from Charles River Laboratories International, Inc. (Wilmington, MA, USA). The ZDF is an obese model, a substrain of the outbreed Zucker rats in the laboratory of Dr. Walter Shaw at Eli Lilly Research Laboratories in Indianapolis, which enables us to conduct research with them in the field of diabetology and obesity. All of the methods used in this present study were approved by the local Ethics Committee of University of Debrecen (25/2013DEMÁB; 29.04.2014). The animals received proper care and were caged on the basis of the “Principles of Laboratory Animal Care” by EU Directive 2010/63/EU. They were fed with Purina 5008 diet and had free access to water.

The present study began with two weeks of acclimatization. After this period of time, the animals were randomly divided into three groups, a healthy control group (LEAN, *n* = 10), a diseased group modelled by ZDF rats (*n* = 15), and a group of ZDF rats that were administered BGP-15 (*n* = 15). Each animal received the same dose (10 mg/kg/daily) of either vehicle (Hydroxyethyl cellulose: distilled water; 1:5 mixture) or drug candidate BGP-15 for 52 weeks by oral gavage. BGP-15 was purchased from Sigma-Aldrich-Merck KGaA (Darmstadt, Germany). The dosage was defined based on previous studies [13].

### 2.2. Chemicals

All chemicals, reagents, and buffer solutions used for isolation, Western blot, Cytochrome c oxidase and ATP synthase assay were obtained from Sigma-Aldrich–Merck KGaA (Darmstadt, Germany), and Abcam Plc. (Cambridge, UK). Ca^2+^-Containing Modified Krebs Solution (in mmol/L): NaCl: 118, KCl: 4.7, CaCl_2_: 2.5, NaH_2_PO_4_: 1, MgCl_2_: 1.2, NaHCO_3_: 24.9, glucose: 11.5, and ascorbic acid: 0.1, dissolved in redistilled water. Ca^2+^-Containing Modified Krebs buffer was used to wash isolated organs for further studies.

### 2.3. Echocardiographic Studies

Transthoracic echocardiography of rats was carried out by a Vivid E9 sonographic equipped with an i13L linear-array probe (GE Healthcare, New York, NY, USA). All of the animals were anaesthetized intramuscularly with ketamine/xylazine combination (75/5 mg/kg, respectively), then the chest hair was removed and animals were placed on a heated table. Data acquisition was performed in 2D-, M-, and Doppler modes, from parasternal short (SAX)- and long axis (PLAX, respectively), as well as apical axis (APLAX). Cardiac function was evaluated in accordance with the guideline of American Society of Echocardiography [15,16]. Wall thickness and chamber diameters were measured in M-mode, at the mid-papillary level. Diameter of the left atrium (LA) was normalized to aortic root (Ao) diameter (LA/Ao). Left ventricle internal diameter in diastole (LVIDd) and in systole (LVIDs), wall thickness of the septal, and lateral walls were measured by standard methods. Fractional shortening (FS), Ejection fraction (EF) mitral, and tricuspid plane systolic excursion (MAPSE and TAPSE, respectively) were calculated from M-mode traces. Diastolic function was assessed by Doppler (pulsed wave, PW) and Tissue Doppler imaging (TDI) from apical 4-chamber views. The ratio of peak early (E) and atrial (A) transmitral flow velocities were assessed (E/A ratio), and deceleration time of the E wave (DecT) was measured. Left ventricle outflow tract (LVOT) parameters (velocity: V; and pressure gradient: PG) were evaluated using PW mode from the apical 5-chamber view. Early (e’) and atrial (a’) diastolic myocardial relaxation velocity waves, systolic myocardial velocity (s’), mitral valve closure to opening time (MCOT), isovolumic contraction time (IVCT), ejection time (ET), and isovolumic relaxation time (IVRT) were defined in tissue Doppler (TDI) imaging at the septal and mitral annulus. The ratio of E/e’ was calculated offline. Myocardial performance index (Tei-index) was determined as a sum of the isovolumic contraction and relaxation times (ICT and IRT) divided by the ejection time (ET). Images were analysed by a single reader in a blinded fashion using the EchoPAC PC software ver. 112, (GE Healthcare, New York, NY, USA). To assess the quality of measurements, three cardiac cycles were averaged for each parameter at each animal, and data are presented as mean ± SD.

### 2.4. Analysis of Serum Parameters

Blood samples were collected in Vacutainer Plast SSTII tubes (BD Vacutainer, Bergen County, NJ, USA), using a 23-gauge needle from the lateral saphenous veins of each animal after 12 h fasting. The samples were handled aseptically to minimize haemolytic activity. Insulin was measured on the Liaison XL DiaSorin platform (DiaSorin Inc., Stillwater, MN, USA). All other serum parameters were measured on the Roche Cobas Integrated platform (Roche Diagnostics GmbH, Mannheim, Germany). Lipid parameters included total cholesterol, low-density lipoprotein (LDLc), and high-density lipoprotein (HDLc). Specific cardiac biomarkers were troponin T, creatine kinase MB isoform (CK-MB), and lactate dehydrogenase (LDH). For blood sugar levels, related measurements blood was collected from the tail vein. Fasting blood glucose concentration was calculated by an Accu-Check glucose meter (Roche Diagnostics, Mannheim, Germany). For quantifying insulin resistance and ß-cell function, the homeostasis model assessment (HOMA) was used for calculation with the following formulas, where HOMA-IR corresponds to insulin resistance and HOMA-B to ß-cell function: HOMA-IR: fasting glucose × insulin/22,5; HOMA-B: 20 × insulin/fasting glucose–3,5 [13].

### 2.5. Morphometry

At the endpoint of the study, all animals were weighed and anaesthetized deeply with ketamine-xylazine (75/5 mg/kg) intramuscular injection, and thoracotomy was performed. Organ samples (heart, lung, kidney, liver, and right tibia) were excised, washed in modified Krebs buffer, and weighed using a milligram scale. Left ventricle and septum of the hearts were isolated, and weights were measured and normalized to tibia length. After weighing, kidney and lung samples were kept overnight at 60 °C and wet-to-dry tissue ratios were determined. Cardiac samples that remained were stored in 4% formalin solution or were rapidly deep-frozen in N2 and stored at −80 °C for further analyses.

### 2.6. Histology

Hearts were dissected transversely at mid-LV level, and samples were fixed for 24 h in 4% neutral buffered formalin (pH = 7.4). From the formalin-fixed, paraffin-embedded (FFPE) blocks, 4 μm thick sections were created and stained with haematoxylin-eosin (HE). To estimate the degree of interstitial fibrosis Masson’s trichrome staining was performed too on the sections, based on the protocol provided by the manufacturer (Sigma-Aldrich Co., St. Louis, MO, USA). Image acquisition was performed using Nikon Eclipse 80i microscope. On the other hand, in order to determine the extent of each cardiomyocyte, transnuclear transversal area was measured. Seven animals from the control group, six from the ZDF group, and seven from the ZDF+BGP-15 group were selected, and 11 sections from the control and ZDF+BGP-15 groups and 12 sections from the ZDF group were created from their dissected heart. One hundred longitudinally oriented cardiomyocytes from LV were examined on each section (magnification 100×), and the diameters at the transnuclear position were defined with Image J Software. According to statistical analysis the mean values of 100 measurements represent one section, and the mean values of sections originating from the same animal were adjusted (mean of the mean).

### 2.7. Western Blot

To identify proteins from myocardial tissue of the left ventricle, Western blot technique was used. At the beginning of the procedure, deep frozen samples (300 mg at –80 °C) were treated with liquid nitrogen and they were milled for protein extraction. The powder arising from the tissue of the heart was homogenized by a Poltroon-homogenizer (IKA-WERKE, Staufen, Germany) in 800 µL Buffer (25 mM Tris-HCl, pH = 8, 25 mMNaCl, 4 mMNa-orthovanadate, 10 mMNaF, 10 mMNa-pyrophosphate, 10 nM okadaic acid, 0.5 mM EDTA, 1 mM PMSF and protease inhibitor cocktail (Sigma-Aldrich, St. Louis, MO, USA)). Total protein concentration was determined from the supernatant after centrifugation (at 10,000× *g* for 20 min). For this part of the analysis, Bicinchoninic Acid (BCA) Protein Assay Kit (QuantiPro™ BCA Assay Kit, Sigma-Aldrich-Merck KGaA, Darmstadt, Germany) was used. Samples containing the proper amount of total protein (20 µg) were separated by Gel electrophoresis (using 10, 12 or 18% gel), since this method enables us to divide proteins examined according to their molecular weight. The procedure was conducted at 40 mA for 100–120 min. After SDS-Polyacrylamide gel electrophoresis, electro-blotting (at 25 V for 90 min) made it possible to transfer proteins onto a nitrocellulose membrane (GE Healthcare, New York, NY, USA). After a period of blocking (1 h at room temperature) in TBS-T, which contained 3% BSA the membranes were incubated overnight with the following antibodies: anti-Beta actin (Beta actin), anti-superoxide-dismutase 1 (SOD1), anti-superoxide-dismutase 2 (SOD2), anti-heat-shock-protein 32 (HSP32), anti-Ubiquinol-Cytochrome-C-Reductase-Binding-Protein (UQCRB), anti-cyclooxygenase 2(MTCO), and anti-ATP-synthase (ATPS). All of the primary and secondary antibodies (anti-Rabbit and anti-Mouse) were purchased from Sigma-Aldrich (Sigma-Aldrich-Merck KGaA) and Abcam (Abcam Plc., Cambridge, UK) and were applied after the recommendation of the manufacturer. The second incubation was conducted with secondary antibodies followed by labelling with horseradish peroxidase. For detection of the proteins, enhanced chemiluminescence reagent was used. Then, the membranes were scanned with a C-Digit© blot scanner with Image Studio Digits ver. 5.2. software (LI-COR Inc., Lincoln, NE, USA). In all cases, the background was normalized, and a standardization was conducted to a housekeeping protein (Beta-actin); then, the statistical analysis was carried out from the average of three independent experiment.

### 2.8. Cytochrome C Oxidase Enzyme Activity Microplate Assay

To measure the activity of Cytochrome c oxidase enzyme (CYTOCOX), an optimized colorimetric assay was used. The method was carried out according to the instructions of manufacturer. Briefly, the Kit is a colorimetric assay based on the observation of the decrease in absorbance of Ferro cytochrome c measured at 550 nm, which is caused by its oxidation to Ferri cytochrome c by cytochrome c oxidase (Sigma-Aldrich-Merck KGaA, Darmstadt, Germany).

### 2.9. ATP Synthase Enzyme Activity Microplate Assay

For examination of the activity of complex V, ATP synthase Enzyme Activity Microplate Assay kit was used, which was obtained from Abcam (Abcam Plc., Cambridge, UK). After preparing samples from left ventricle myocardial tissue following the instruction of the manufacturer, the plates included in the kit were loaded and incubated at room temperature for 3 h. The previously prepared samples were rinsed with solution 1 used during the preparation, lipid mix, and reagent mix were added as recommended. The activity of the enzyme was measured on OD340 in 1 min intervals for 1–2 h.

### 2.10. Statistical Analyses

All data are presented as mean ± standard deviation (SD). D’Agostino and Pearson normality test was used to estimate normal distribution. Then, statistical analysis was performed using one-way analysis of variance (ANOVA) followed by Tukey post hoc analysis (only if F achieved *p* < 0.05, normality test was passed, and there was no significant variance inhomogeneity), or Kruskal–Wallis test followed by Dunn’s post-test (when normality test was not passed). Analyses were carried out using GraphPad Prism software for Windows, version 7.00 (GraphPad Software Inc., La Jolla, CA, USA). *p* < 0.05 was considered as statistically significant.

## 3. Results

### 3.1. BGP-15 Prevented Ageing and Type 2 Diabetes Mellitus-Associated Cardiac Dysfunction In Vivo

ZDF rats showed signs of both systolic and diastolic dysfunction. Ejection fraction of ZDF group decreased in comparison to LEAN (66.22 ± 1.949 vs. 80.63 ± 1.711%, *p* < 0.0001), similarly to fractional shortening (32.67 ± 1.374 vs. 44.75 ± 1.78%, *p* < 0.0001). MAPSE was diminished in ZDF group compared to LEAN (1.777 ± 0.0577 mm vs. 2.335 ± 0.098, *p* = 0.0022) (Figure 1d). Systolic mitral annular velocity (s’) decreased in the ZDF rats vs. LEAN (30.5 ± 2.885 vs. 41.88 ± 2.248 mm/s, *p* = 0.0105). Right ventricle systolic function, described by TAPSE, was deteriorated in the ZDF group vs. LEAN (2.281 ± 0.0167 vs. 3.596 ± 0.0242 mm, *p* = 0.0001). ZDF rats showed impaired diastolic function compared to controls. LA/Ao increased in ZDF (1.283 ± 0.052 vs. 0.9697 ± 0.0351, *p* < 0.0001), e’/a’ ratio decreased (0.776 ± 0.054 vs. 1.298 ± 0.65, *p* = 0.0495), and E/e’ ratio increased (26.03 ± 2.525 vs. 18.20 ± 1.246, *p* = 0.0088) (Figure 1h). IVRT lengthened in the ZDF group compared to LEAN (55.0 ± 3.674 vs. 33.13 ± 1.093, *p* < 0.0001) and Tei-index worsened (0.643 ± 0.0297 vs. 0.5091 ± 0.0164) (Figure 1e). E/A ratio and DecT did not show significant changes; however, we note that the animals in the ZDF group showed either extremely high or extremely low E/A values, and together with the above parameters (E/e’, IVRT), we showed that all animals in the ZDF group were in different stages of DD at the age of 52 weeks. Echocardiographic parameters were significantly improved in the BGP-15 treated group in comparison to ZDF. Systolic function of ZDF+BGP-15 rats were preserved in comparison to ZDF, as treated rats showed improved EF: (82.20 ± 0.509, *p* < 0.0001), FS (45.93 ± 0.502, *p* < 0.0001), MAPSE (2.354 ± 0.1089, *p* = 0.0007), and TAPSE (3.658 ± 0.081, *p* < 0.0001) values (Figure 1c,d). Left atrial enlargement was attenuated in the BGP-15 treated group vs. ZDF (LA/Ao: 0.951 ± 0.021, *p* < 0.0001), E/A ratio was normal (1.598 ± 0.073), E/e’ was improved (18.01 ± 1.018, *p* = 0.0022), IVRT was normalized (44.6 ± 1.704, *p* = 0.0068), and Tei-index improved (0.516 ± 0.016, *p* = 0.0003). No significant differences were found in LVOT parameters (outflow velocity and pressure gradients).

### 3.2. Severely Diabetic Rats Demonstrated Decreased Body Weight

Throughout the time of the study, the bodyweight of the animals was measured monthly after the sixth month. At the endpoint, the weight of the animals in the control group was almost twice as much as in the other groups, and the difference was significant (Control vs. ZDF and Control vs. ZDF+BGP-15 *p* < 0.0001). The heart weight was higher in the control group, but the difference was only significant between the control and the treated group (Control vs. ZDF+BGP *p* = 0.031). There was no significant difference in the heart weight–tibial length quotient, but for both the bodyweight–tibial length and heart weight–bodyweight quotient, there was a recognizable difference between the control, the diseased, and the treated group (BW/TL: Control vs. ZDF and Control vs. ZDF+BGP *p* < 0.0001; HW/BW) (Table 1).

### 3.3. Ageing and T2DM Elevates Serum Parameters

Values of serum lipid parameters and cardiac biomarkers are shown in Table 2. The level of total cholesterol was the highest in the ZDF group, but there was a significant difference observable between the treated and control groups as well (Control vs. ZDF *p* = 0.0018; Control vs. ZDF+BGP-15 *p* = 0.0484). The results for the LDL levels are similar; the highest levels were measured in the ZDF group (Control vs. ZDF *p* = 0.0028; ZDF vs. ZDF+BGP-15 *p* = 0.043). There was no significant difference between the HDL levels. Creatinine levels ought to be similar to the total cholesterol and LDL levels (Control vs. ZDF *p* = 0.0249; Control vs. ZDF+BGP-15 *p* = 0.0348). The creatine-kinase level of ZDF rats was significantly higher than measured in the treated group (ZDF vs. ZDF+BGP-15 *p* = 0.04). Although the levels of LDH and Troponin T were higher in the diseased group than either in the control or the treated group, but the difference were not showing any significance.

### 3.4. BGP-15 Has Mild Influence on Glucose Homeostasis

Fasting glucose levels were higher in both the ZDF and the treated groups, but the difference was not significant between the control and ZDF+BGP 15 groups (Control vs. ZDF *p* = 0.0005; ZDF vs. ZDF+BGP-15 *p* = 0.0148). The insulin levels resulted in a significant difference in the same groups (Control vs. ZDF *p* = 0.0076; ZDF vs. ZDF+BGP-15 *p* = 0.0076). In the point of HOMA-IR a significant difference was observable between the control and treated groups (Control vs. ZDF+ZDFBGP-15 *p* = 0.0259). According to HOMA-B index the ß-cell function was in the ZDF group the worst, the difference was significant in relation to the other two groups (Control vs. ZDF *p* = 0.0016; ZDF vs. ZDF+BGP-15 *p* = 0.0146) (Table 3).

### 3.5. BPG-15 Decrease Myocardial Hypertrophy in Ageing Zucker Diabetic Fatty Rats

A significant difference was observed in point of the cardiomyocyte diameters between the control and diseased group (*p* = 0.041). Although the extent of cardiomyocytes ought to be higher in the diseased group (ZDF) then in than in the treated group (ZDF+BGP-15), the difference was not significant (*p* = 0.213) (Figure 2).

### 3.6. BGP-15 Improves Mitochondrial Function in Ageing Zucker Diabetic Fatty Rat Hearts

In an attempt to further confirm the increase in mitochondrial oxidative capacity, we analysed the subunits of the electron transport chain by Western blotting. Most of the subunits of oxidative phosphorylation were increased in mitochondria from ZDF BGP-15-treated rats compared to the ZDF non-treated group. The relative expression of NADH dehydrogenase (NDFUS4) was the highest in the control group, and there was a significant difference observable between the diseased and the healthy group (Control vs. ZDF *p* = 0.0111). A high relative expression was recognized in Cytochrome c oxidase (MTCO) in BGP-15 treated groups compared to ZDF and LEAN groups (Control vs. ZDF+BGP-15 *p* = 0.0007; ZDF vs. ZDF+BGP-15 *p* = 0.0007). Concerning the relative expression of ATPS, similar results were inspected (Control vs. ZDF+BGP-15 *p* = 0.0031; ZDF vs. ZDF+BGP-15 *p* = 0.002). Further analysis of the enzymatic activity of the mitochondrial ETC complexes (Complex IV-V) exhibited the same tendencies (Figure 3).

### 3.7. BGP-15 Treatment Induces MTCO and ATP Synthase Activity in Ageing Zucker Diabetic Fatty Rats

Cardiac muscle mitochondrial content was determined via MTCO (aka CYTOCOX) enzyme activity analysis, an established marker of mitochondrial biogenesis. A remarkable increase was observed in BGP-15 treated group in MTCO activity compared with ZDF group (209.8 ± 150.3 vs. 54.9 ± 57.11; *p* = 0.0445). In addition, BGP-15 group showed no significant differences compared to LEAN group (209.8 ± 150.3 vs. 100 ± 68.53, *p* = 0.1773). Moreover, ATP synthase activity was significantly increased in BGP-15 group compared to ZDF as well (169.1 ± 54.23 vs. 79.84 ± 67.42, *p* = 0.0201) (Figure 3).

### 3.8. Ageing and T2DM Exerts an Antioxidant Response on ZDF Rat Hearts by HO-1 Dependent Activation

To assess the antioxidant response in cardiac cytoplasm and mitochondria, we examined HO-1 (aka Hsp32), SOD1 (aka Cu/ZnSOD) and SOD2 (aka MnSOD) expression by Western blot technique as well. SOD1 and SOD2 proteins expression did not showed any differences among groups, in contrast HO-1 level was significantly elevated in ZDF group, compared with LEAN and ZDF BGP-15 groups (Control vs. ZDF *p* = 0.0001; ZDF vs. ZDF+BGP-15 *p* = 0.0011) (Figure 4).

## 4. Discussion

### 4.1. General Characteristics of the ZDF Rats: Glucose and Lipid Homeostasis

It is a well-known phenomenon that diabetes mellitus (DM) increases the risk of cardiac dysfunction independently of coronary artery disease or even blood pressure. The main problem is that uncontrolled diabetes acts in the affected non-healthy patient as a sneaking silent killer; therefore, it is crucial to reveal all signal transduction pathways of its pathomechanism. In addition, having a mutation in leptin receptor, like in Zucker diabetic fatty (ZDF) rat is a well-established animal model of type 2 DM. Regarding cardiac dysfunction, earlier studies reported highly divergent results in relation to depressed diastolic function alone or in combination with systolic LV function [17,18]. Since the association between increasing age and T2DM is strong, studying aged rats is more relevant to the clinical situation. To dissolve this discrepancy, in our present study, we conducted multiple investigations and only terminated them after 52 weeks. Taking into account that ZDF rats have limited lifespan, the animals involved in this study should be considered as aged animals. Additionally, to the best of our knowledge, this long-term follow-up and treatment is unique so far [19]. Based on our blood parameters of the serum, morphometric examination, aged ZDF rats showed representative features of severe and prolonged T2DM. T2DM was exemplified by strongly elevated fasting glucose, low insulin level, and extremely low body weight compared to the healthy LEAN group [20]. Considering the glucose-metabolism-related values, in BGP-15-treated ZDF animals, only insulin resistance was observed due to the higher insulin level compared to the ZDF group. These findings suggest that even though BGP-15 therapy fails to reduce blood sugar level effectively, it clearly slows down the progression of T2DM. The ZDF rat fed with lipid-rich diet closely mimics human adult onset of metabolic syndrome with age-dependent evolution from insulin resistance to type-II diabetes mellitus, hypertriglyceridemia, and hypercholesterolemia. In our model, all of the lipid homeostasis markers were elevated in both ZDF groups, although LDLc was significantly lower in BGP-15 treated group compared to ZDF rats. To our knowledge, this is the first study describing BGP-15 LDL cholesterol lowering effect. Peterson et al. define, that serum lipids rises after 22–42 weeks of age, when insulin resistance develops into severe T2DM [21]. These data support the hypothesis that BGP-15 delays the onset of metabolic syndrome in ZDF rats.

### 4.2. Assessment of Cardiac Dysfunction and Remodelling

During haemodynamic measurements, signs of diabetes-induced myocardial structural and metabolic remodelling were clearly presented. In this current study, we observed that severe T2DM in ZDF rats leads to significantly depressed cardiac haemodynamic performance with depressed LV diastolic and systolic function as well. A study on 45-week-old ZDF rats even reported similar results. Daniels et al. observed that long-term severe diabetes and ageing leads to mild impairment of diastolic LV function, whereas systolic function was well preserved [19]. Interestingly, in our study, animals in the untreated ZDF group were in different stages of diastolic dysfunction with extremely low and extremely high E/A values, based on the diastolic dysfunction stages/grades. The cardioprotective effects of BGP-15 were seen even in diastolic as well as in systolic parameters. This finding is in line with investigations of Spontaneously Hypertensive Rat (SHR) rats. BGP-15 treatment results considerably decrease the worsening in systolic (EF) and in diastolic (E/E’) LV function. Moreover, only mild LV hypertrophy was seen in the BGP-15-treated group [22]. In contrast, our findings showed preserved cardiomyocyte diameters in BGP-15 treated group compared to LEAN healthy control group, and it was only the ZDF group that showed significant difference in terms of cardiac hypertrophy. Recent findings suggest a so-called reverse remodelling phenomenon induced by BGP-15 long-term treatment. Cardiac biomarkers (e.g., CK including CK-MB, LDH, Troponin T) which leak out from damaged tissues into the blood stream serve as diagnostic markers of myocardial injury. Some of these, e.g., CK, LDH, and Troponin T, showed slight elevation in the ZDF group compared to the control group; nevertheless, they did not reach the level of significance. In addition, creatinine levels were elevated significantly in both ZDF group, but only CK level was significantly decreased in BPG15 compared to ZDF rats. These results indicate the obvious detrimental effects of diabetes on cardiac tissues [23]. Additionally, the elevated level of these markers correlate with the disease severity and with the cardiac dysfunction detected with the help of echocardiographic measurements.

### 4.3. ROS Capacity

To our knowledge, increased intramitochondrial haem and subsequent ROS generation may be the driving force for mobilizing HO-1 in mitochondria. Under oxidative injury provoked by diabetes or ageing in some tissues, haem-derived Fe and CO may exacerbate intracellular oxidative stress and cellular injury by promoting ROS generation in mitochondria. HO-1 enzyme is a well-known cytoprotective stress protein (Hsp32); it provides defence against oxidative stress by catalysing the degradation of pro-oxidant haem to biliverdin and bilirubin. At the same time, acting as a double-edged sword, the overexpression of HO-1 promotes mitochondrial sequestration and macroautophagy [24,25,26,27]. In turn, the overexpression of HO-1 drives the overproduction of CO, which leads to tissue hypoxia and direct CO-mediated cell damage. Due to the short half-life, mitochondria and especially complex IV (also known as cytochrome c oxidase) are crucial members of the electron transport chain and seem to be the directly inhibited target by CO [28,29]. In our ageing ZDF model, a noticeable HO-1 overexpression was observed, supporting the abovementioned correlation with oxidative stress and diabetes. However, mitochondrial and cytoplasmic superoxide dismutases (SOD1 and SOD2) were also detected by Western blot technique, but significant elevation in protein expression in our hyperglycaemic tissue was not seen. The limitation of this study is that SOD activity was not measured, although based on the literature, a decreased activity is expressed in patients with DM. The inactivity of SOD might be due to the hyperglycaemia, which leads to glucose auto oxidation and non-enzymatic glycosylation of numerous proteins, such as SOD [30].

### 4.4. Upregulation of Cardiac ATP Synthase and Cytochrome C Oxidase

As was noted above, the heart has a high and continuous demand for oxidative metabolism to maintain ATP production. Accordingly, being the centre of fatty acid and glucose metabolism, a cardiomyocyte has a relatively high number of mitochondria compared to other tissues. In physiologic demand, the heart preferentially uses fatty acids, but in pathologic conditions, such as hypertrophy, heart failure, and myocardial infarction, the fuel preference is switched toward glucose oxidation with less oxygen consumption. This means that a diabetic heart has an increased rate of fatty acid oxidation and decreased rate of glucose oxidation. In addition, development of advanced glycation end (AGE) products and the modulation of NADPH oxidase and Superoxide contribute an increased ROS level as well. Due to their short half-life, mitochondrial ROS production may play important role regarding the mitochondrial damage during diabetes [31,32]. Considering the above, metabolic remodelling and a defect in mitochondrial bioenergetics lead to cardiac dysfunction in diabetes. In our study, the decrease in mitochondrial function was revealed by a decline in oxidative phosphorylation at the level of Complex I and II. Under normal conditions, both complexes generate NADH and Dihydroflavine-adenine dinucleotide (FADH_2_) during the oxidization of substrates, pass electrons to coenzyme Q, and further reduce Complex III, cytochrome c, and Complex IV [32]. Our study identifies the decreased ETC function on the level of protein expression of Complex I, IV, and V in diabetes. Defects of Complex IV and V were proved by decreased activity of enzymes as well. We showed that LV dysfunction and remodelling are associated with Complex I, IV, and V defects. At the same time, we found a dramatic increase in the activity of Cytochrome c oxidase and ATP synthase with the treatment of aged rats that suffered from T2DM with the pharmacological agent BGP-15.

### 4.5. Translational Aspects and Future Directions

According to these findings, we strongly hypothesize that BGP-15 protects mitochondrial function against diabetes- and ageing-derived oxidative stress. Long-term treatment with this nicotinic acid (Vitamin B3) derivative delays the onset of severe T2DM and metabolic changes. Based on our and previous results, the cardioprotective effect of BGP-15 is indisputable. In the future, it would be worth considering the BGP-15 drug candidate as a NAD precursor since it may provide sound basis for a promising strategy in the treatment of multiple pathologic conditions related to NAD deficiency.

## 5. Conclusions

To the best of our knowledge, this is the first study that demonstrates BGP-15′s protective effect on cardiac mitochondrial respiration in Zucker Diabetic Fatty Rat model at the age of 52 weeks. Due to a strong association between increasing age and T2DM, examination of aged rats is more relevant to the clinical situation. At the end of our study, age-dependent aggravation of diabetes, cardiac dysfunction, and cardiac remodelling were all clearly seen in diabetic rats, although in BPG15-treated animals, this metabolic features did not reach this life-threatening level. We assume that as a nicotinic acid derivative, BGP-15 may act as a NAD precursor and have all the beneficial effects on mitochondrial energy metabolism on subcellular level. In spite of our statement, further investigations are needed, and these observations undoubtedly place BPG-15 into a new focus of therapeutic approach.

## Figures and Tables

**Figure 1 pharmaceutics-14-00226-f001:**
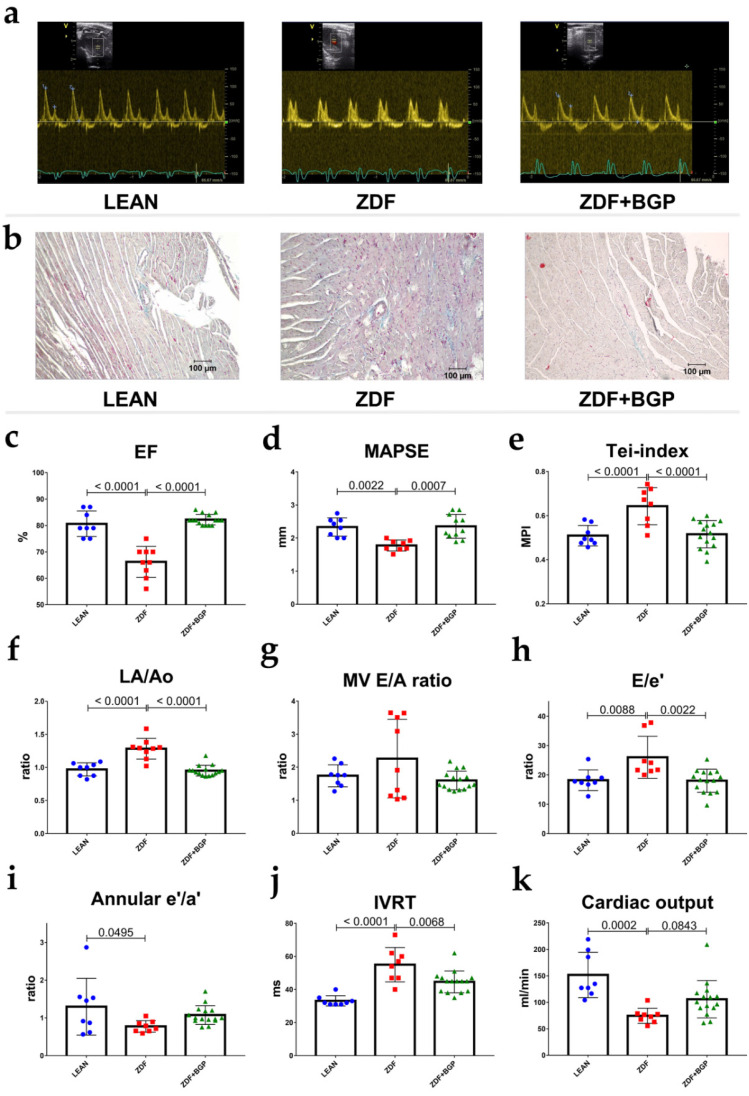
Echocardiographic parameters of rats at the endpoint of the study. BGP-15 treatment improved the echocardiographic parameters of aged ZDF rats. (**a**) Representative Doppler traces of the mitral inflow (E and A waves) in LEAN, ZDF, and ZDF+BGP groups, respectively. (**b**) Histological samples of the dissected heart tissues stained with Masson’s trichrome in LEAN, ZDF, and ZDF+BGP groups, respectively. Fibrotic tissue dyes with blue colour. (**c**) Ejection fraction (EF) in the three endpoints groups. EF was deteriorated in the ZDF, but was normal in the ZDF+BGP group. (**d**) Mitral annular plain systolic excursion (MAPSE) improved in the treated animals. (**e**) Tei-index (Myocardial Performance Index, MPI) worsened in the ZDF but improved in the ZDF+BGP group. (**f**) Left atrial enlargement (LA/Ao) improved in the treated animals. (**g**) Mitral valve inflow E/A ratio (MV E/A) in the endpoint groups. (**h**) E/e’ ratio, indicating left ventricle filling pressure was elevated in the ZDF but was restored in the ZDF+BGP group. (**i**) Septal annular wall motion e’/a’ ratio worsened in the ZDF animals. (**j**) Isovolumic relaxation time (IVRT) lengthened in the ZDF, but was restored in the ZDF+BGP group. (**k**) Cardiac output (calculated by echocardiography) reduced in the ZDF but increased in the treated group. Normal data distribution, data presented as mean ± SD, one-way ANOVA, and Tukey post-test.

**Figure 2 pharmaceutics-14-00226-f002:**
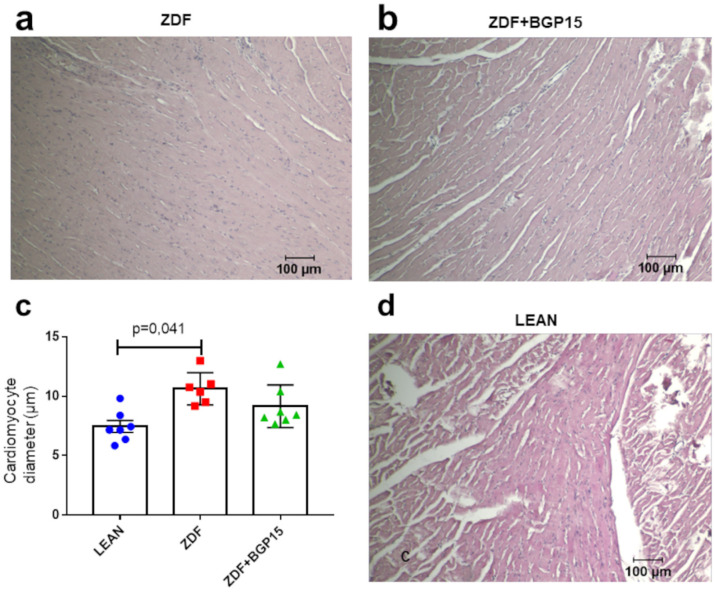
Histological samples stained with haematoxylin and eosin and their quantification. (**a**) Histological sample from the diseased group (ZDF), (**b**) histological sample from the BGP-15-treated ZDF group, (**c**) cardiomyocyte diameters among groups. All data are presented as mean ± standard deviation (SD, (**d**) histological sample from the control (LEAN) group, and). *p* < 0.05. Statistical analysis was carried out by GraphPad Prism 7.00: D’Agostino and Pearson normality test was used to estimate Gaussian distribution, and then data were analysed with ordinary one-way ANOVA or Kruskal–Wallis test. The groups were matched by Tukey’s multiple comparisons test.

**Figure 3 pharmaceutics-14-00226-f003:**
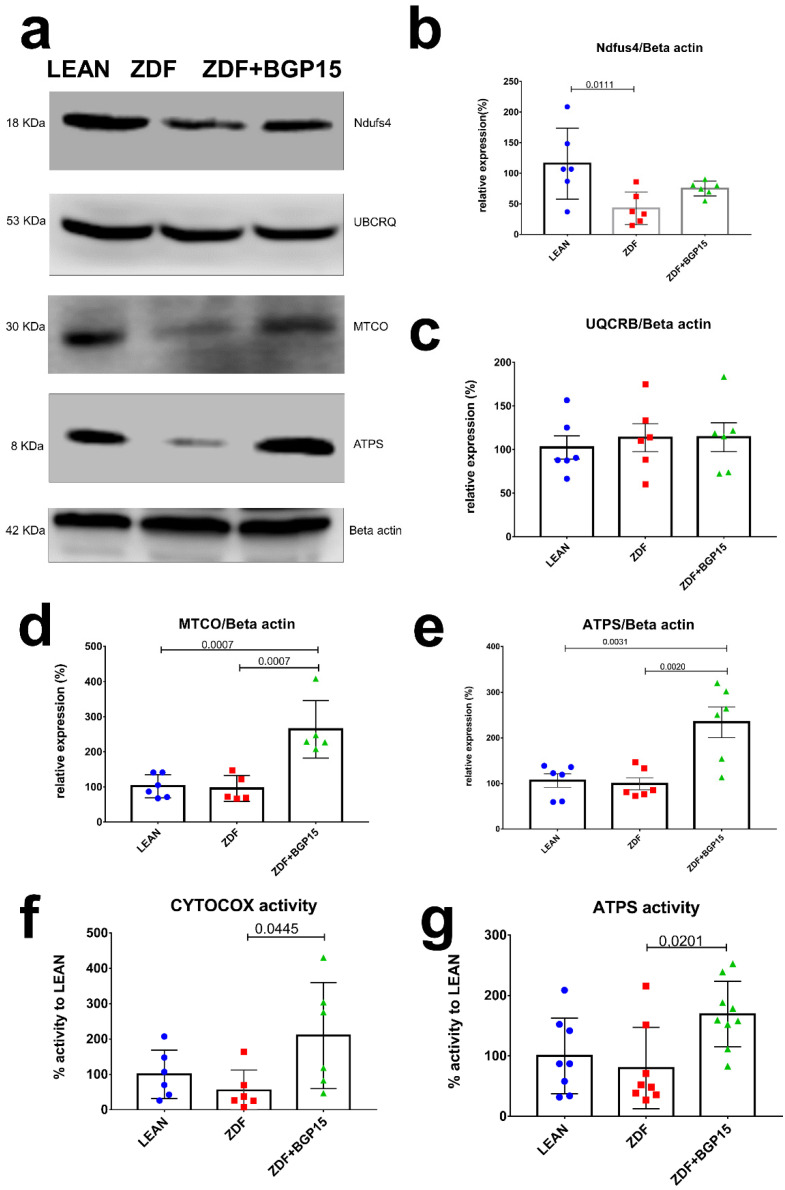
Protein expression and activity of electron transport chain complexes. BGP-15 treatment protects mitochondrial function against diabetes-induced oxidative stress in ageing ZDF rat myocardium. (**a**) A representative Western blot for electric transport chain complex I, III, IV, and V in ZDF rat myocardium treated with BGP-15. (**b**–**e**) Quantitative band density analyses of the ETC proteins normalized to Beta actin and expressed in the percentage of LEAN (that was considered 100%). (**f**,**g**) Graphs show MTCO (aka CYTOCOX) and ATP synthase enzymes activity analysed by microplate assays. Results are expressed as percentage of LEAN (which was considered 100%). All data are presented as the average outcome in a group (mean) ± standard deviation (SD). Statistical analysis was carried out by GraphPad Prism 7.00: D’Agostino and Pearson normality test was used to estimate Gaussian distribution, and then data were analysed with ordinary one-way ANOVA or Kruskal–Wallis test. The groups were matched by Tukey’s multiple comparisons test.

**Figure 4 pharmaceutics-14-00226-f004:**
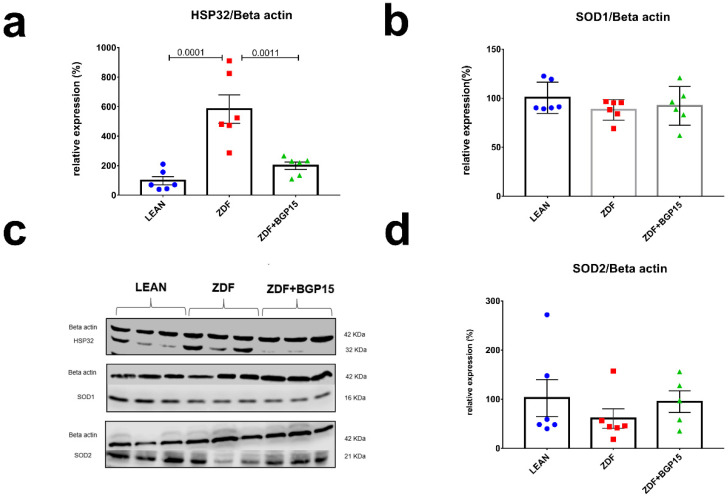
Expressions of antioxidant proteins from rat myocardium. HO-1 protein expression elevated significantly to diabetes induced oxidative stress in aged rats. (**a**,**b**,**d**) quantitative band density analyses of HO-1, SOD1, and SOD2 proteins normalized to Beta actin and expressed in the percentage of LEAN (that was considered 100%). (**c**) A representative Western blot for Beta actin, Hsp32, SOD1, and SOD2 in ZDF rat myocardium treated with BGP-15. All data are presented as the average outcome in a group (mean) ± standard deviation (SD). Statistical analysis was carried out by GraphPad Prism 7.00: D’Agostino and Pearson normality test was used to estimate Gaussian distribution than data were analysed with ordinary one-way ANOVA or Kruskal–Wallis test. The groups were matched by Tukey’s multiple comparisons test.

**Table 1 pharmaceutics-14-00226-t001:** Bodyweight, heart weight (BW, HW) to tibia length (TL) ratios, lung and kidney wet (w) to dry (d) tissue ratios at the endpoint of the study.

		LEAN	ZDF	ZDF+BGP-15
N		8	7	9
Parameter	Unit	Mean ± SD	Mean ± SD	Mean ± SD
BW	g	453.6 ± 14.76	# 279.3 ± 53.76	# 279.9 ± 40.84
HW	g	1.221 ± 0.15	1.09 ± 0.12	# 1.02 ± 0.18
HW/TL	g/cm	0.25 ± 0.05	0.24 ± 0.03	0.23 ± 0.04
BW/TL	g/cm	91.14 ± 6.79	# 60.13 ± 11.2	# 58.86 ± 9.59
HW/BW	g/kg	2.69 ± 0.36	# 4 ± 0.7	# 3.76 ± 0.95
lung w/d		0.19 ± 0.02	0.19 ± 0.01	0.18 ± 0.01
kidney w/d		0.26 ± 0.03	0.23 ± 0.07	0.25 ± 0.03

Weight of the animals and organs in the endpoint of the study. All data are presented as mean ± standard deviation (SD). *p* < 0.05. #: significant difference compared to control group. Statistical analysis was carried out by GraphPad Prism 7.00: D’Agostino and Pearson normality test was used to estimate Gaussian distribution than data were analysed with ordinary one-way ANOVA or Kruskal–Wallis test. The groups were matched by Tukey’s multiple comparisons test.

**Table 2 pharmaceutics-14-00226-t002:** Values of cardiac- and lipid homeostasis biomarkers.

		LEAN	ZDF	ZDF+BGP-15
N		8	6	8
Parameter	Unit	Mean ± SD	Mean ± SD	Mean ± SD
Total Cholesterol	mmol/L	2.19 ± 0.36	# 8.09 ± 4.21	# 5.62 ± 2.59
LDLc	mmol/L	0.23 ± 0.08	# 1.08 ± 0.68	* 0.51 ± 0.33
HDLc	mmol/L	0.49 ± 0.15	1.16 ± 0.68	0.69 ± 0.58
Creatinine	µmol/L	26.7 ± 12.85	# 141 ± 221.1	# 66.88 ± 53.83
CK	U/L	380.8 ± 304.6	548.8 ± 312.4	* 182.1 ± 124.9
LDH	U/L	1143 ± 1559	1871 ± 1256	1114 ± 856.8
Troponin T	ng/L	3263 ± 2356	4162 ± 3293	3349 ± 2276

All data are presented as mean ± standard deviation (SD). *p* < 0.05. #: significant difference compared to control group; *: significant difference compared to ZDF group. Statistical analysis was carried out by GraphPad Prism 7.0s0: D’Agostino and Pearson normality test was used to estimate Gaussian distribution than data were analysed with ordinary one-way ANOVA or Kruskal–Wallis test. The groups were matched by Tukey’s multiple comparisons test. Units of data are represented according to the recommendation of local Department of Laboratory Medicine.

**Table 3 pharmaceutics-14-00226-t003:** Glucose metabolism related values.

		Control	ZDF	ZDF+BGP-15
N		8	6	8
Parameter	Unit	Mean ± SD	Mean ± SD	Mean ± SD
Glucose	mmol/litre	7.05 ± 0.59	# 18.42 ± 7.93	* 10.74 ± 3.34
Insulin	mU/litre	5.81 ± 1.48	# 2.05 ± 1.02	* 5.81 ± 1.48
HOMA-IR		1.79 ± 0.40	1.87 ± 1.50	# 3.19 ± 0.88
HOMA-B		13.3 ± 5.15	# −0.89 ± 1.54	* 11.92 ± 10.57

All data are presented as mean ± standard deviation (SD). *p* < 0.05. #: significant difference compared to control group; *: significant difference compared to ZDF group. Statistical analysis was carried out by GraphPad Prism 7.00. D’Agostino and Pearson normality test was used to estimate Gaussian distribution than data were analysed with ordinary one-way ANOVA or Kruskal–Wallis test. The groups were matched by Tukey’s multiple comparisons test. Unit of data were represented according to the recommendation of local Department of Laboratory Medicine. HOMA-IR corresponds to homeostasis model assessment of insulin resistance and HOMA-B to homeostasis model assessment of B-cell function.

## Data Availability

The data that support the findings of this study are available from the corresponding author upon reasonable request. Some data may not be made available because of privacy or ethical restrictions.

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
