# Peer review of "Cardioprotective Role of BGP-15 in Ageing Zucker Diabetic Fatty Rat (ZDF) Model: Extended Mitochondrial Longevity"

_pharmaceutics, 2022, doi:10.3390/pharmaceutics14020226_

Round 1

Reviewer 1 Report

The manuscript submitted by Koza, et al indicated that BGP-15 exhibits protection on cardiac mitochondrial respiratory in an aging ZDF rat model. The link of cardioprotection of BGP-15 with mitochondrial longevity sounds interesting.

Major concerns:

  1. Line 88: The animal gender of both ZDF and LEAN rats must be indicated in the manuscript. The animal body weight, strain, and age of LEAN must be stated. 
  2. Lines 98-99: "Each animal received the same dose (10 mg/kg/pro die) of either vehicle or drug candidate BGP-15 for 52 weeks by oral gavage". There are two questions: a. What does "pro die" mean? b. What vehicle did the authors use? 
  3. The rational to use only one dose of BGP-15 is unclear. 
  4. Lines 107-108: CaCI2-free Krebs solution contained (in mmol/L): NaCI: 118; KCI: 4.7. CaCI2: 2.5. Can the authors explain why the CaCI2-free Krebs solution contains CaCI2 in solution? 
  5. Lines 202-203: The resources and names of second antibodies need to be described.   
  6. Line 282: The expression of data in Table 1 is confusing: What does "BW: 453,6" mean? Table 1 must be corrected.
  7. Line 300: Can authors explain the results for increased CK and LDH in ZDF rats? Did author compare the results of CK in ZDF+BGP15 group with Control group? 
  8. Line 328: Figure 2C. What does "micrometer" mean? 

Reviewer 2 Report

I don't any reason not to publish this article as it is presently.

Author Response

please see the attachement

Reviewer 3 Report

I congratulate the authors for the novelty and meticulous work developed using ZDF rats to prove the protective role of BGP-15 on cardiac function which is decreased in Type 2 DM. The manuscript is well organised and the methodology and results properly explained. The graphs are very clean and well presented, I consider very important to show all the experimental data points in the graphs instead of only showing the mean±sem. The interpretation and discusion of the results is also very clear.

Minor changes should be apply to improve the paper. A list is attached in a file:

Abstract

Line 21: Include Type 2 Diabetes Mellitus acronym (T2DM) which is used throughout the text but is not included

Lines 26 and 31: NAD+ (+ superscript)

Line 28: remove coma after generation

Line 29: Extra bracket at the beginning of BGP-15 name, I mean ((3-piperidine….)

Line 35: Acronym HE (hematoxylin-eosin)

Introduction

Review + superscripts in NAD+ and NADP+

The introduction provides enough information about the need of this study as well as the previous work made in the same line. It is well organized

Methods

Line 88: is confusing it sound as if the 30ZDF rats and the 10 LEAN rats are the control. Please rewrite the sentence like this: 30 ZDF rats (8-week-old, 200 g), and 10 LEAN rats (control) were used in the present study

Line 99: (10 mg/kg/pro die) with pro die you mean per day???

Lines 107-109: subscript in chemical formula (eg. MgCl2)

Pay attention to subscripts and superscripts throughout the text (eg. Line 157 Ca2+)

Results

Line 240: You say MAPSE decreases in ZDF rats compared with LEAN but you provide the following figures: ZDF vs LEAN (2.335±0.098 vs. 1.777±0.0577 mm, p=0.0022) which means that ZDF MAPSE value is higher. In line 253 you say this parameter is improved in ZDF+BPG rats, providing a value of 2.354±0.1089. It makes me to think that the first figures (2.335±0.098 vs. 1.777±0.0577 mm, p=0.0022) are opposite, i.e. (1.777±0.0577vs. 2.335±0.098  mm, p=0.0022)

Please write vs. in italics

Line 257: Tei index magnitude seems very high in ZDF+BGP group (80.516±0.016, p=0.0003) compared with the values provided in ZDF and LEAN (0.643±0.0297 vs. 0.5091±0.0164), is that right? Sounds weird

257: Change NO by No, is a negation not nitric oxide

Line 274: Sixth instead of six or otherwise “after six months”

Homogenize the figure legend. In Figures 1 and 3 the subfigures are marked in brackets (a) whereas int figure 2 no

Line 341: I think you have not previously mentioned NDFUS4 in the text (perhaps I am looking for it and is there but I haven´t seen it). Please double check and if so mentioned it with its meaning (in 2.7 section)

Section 3.7.: Please mention that the data described in this section are shown in figure 3f and g

Figure 4C: Show in the WB the conditions. I assume they are LEAN, ZDF and ZDF+BGP, actually I think you are pointing the braces to the legend of 4a graph, but it is not clear enough.

Discussion

Line 393: Rewrite “Being a strong association between increasing age and T2DM”. I think it is not right perhaps “Being strong the association between increasing age and T2DM”

Lines 409 and 419: et al with italic

Line 424: SHR (Spontaneous Hypertensive Rat), please say this

Line 472: FADH2 (subscript)

Author Response

We, the authors highly appreciate your precious time in reviewing our manuscript.  

Round 2

Reviewer 1 Report

This is a revised research manuscript. All concerns have been addressed or corrected by authors except the rationale for single dosage in the presented study. Each table should have a title above the table. Authors should use "dot" rather than "comma" for data presentation.

Author Response

Response to Reviewer 1

Comments and Suggestions for Authors:

Question 1: This is a revised research manuscript. All concerns have been addressed or corrected by authors except the rationale for a single dosage in the presented study. Authors should use "dot" rather than "comma" for data presentation.

Response: The animals received BGP-15 and vehicle treatment daily during the study period by oral gavage.  The dosage was defined based on our previous studies. In the above-mentioned study, Goto-Kakizaki rats received BGP-15 in 10mg/kg daily dose for 12 weeks. This previous work demonstrated that BGP-15 improves diastolic function in the diabetic Goto-Kakizaki rat, independently of antidiabetic action. These results give us a new basis to test the Drug Candidate in another diabetic rat model at the same dose, but for a longer period to see not only diastolic but systolic dysfunctions as well. Priksz at al. (2021) demonstrated in a Hypercholesterolemic Rabbit model, that 10 mg/kg daily dose of BGP-15 for 16 weeks, exert a similar cardioprotective effect on diastolic functions without having any effect on lipid homeostasis. According to these two previous results, we hypothesized, that this daily dosage is beneath the antidiabetic effect, but enough to have a cardioprotective effect.

Question 2: Each table should have a title above the table.

Response: Thank you very much for the reminder. We have made revisions accordingly.  Please, find the changes in the main text.

Question 3: Authors should use "dot" rather than "comma" for data presentation.

Response: Thank you, again. We have made revisions accordingly.  Please, find the changes in the main text.

We have gone through your comments carefully and tried our best to address them one by one. We hope the manuscript has been improved accordingly.
